# Non-Thyroidal Illness Syndrome: A Predictor of Non-Invasive Ventilation Failure and Mortality in Critically Ill Patients

**DOI:** 10.3390/medicina61050927

**Published:** 2025-05-20

**Authors:** Dursun Elmas, Muhammet Cemal Kizilarslanoglu

**Affiliations:** 1Division of Intensive Care, Department of Internal Medicine, University of Health Sciences Turkey, Konya City Hospital, 42020 Konya, Turkey; 2Division of Geriatrics, Department of Internal Medicine, University of Health Sciences Turkey, Konya City Hospital, 42020 Konya, Turkey; drcemalk@yahoo.com.tr

**Keywords:** euthyroid sick syndrome, non-invasive mechanical ventilation, mortality, intensive care unit

## Abstract

***Aim:*** This study aimed to investigate the effect of Euthyroid Sick Syndrome (ESS) on non-invasive mechanical ventilation (NIV) failure and in-hospital mortality. We also examined the impact of prolonged low fT3 levels before intensive care unit (ICU) admission on mortality and NIV failure. ***Methods:*** The study included 386 ICU patients who received NIV. The patients were categorized into two groups: those with ESS and those without (Non-ESS). Additionally, retrospective fT3 levels were examined in the ESS group over 6 months before ICU admission, and 61 patients with prolonged ESS were identified. The primary endpoints of this study (NIV failure and mortality) were compared between study groups. ***Results:*** Of the 386 patients included in the study, comprising 332 survivors and 54 deceased individuals, ESS was observed in 49.0% of the total patient population, with a significantly higher prevalence (*p* = 0.005) among deceased patients (66.7%) compared to survivors (46.1%). Among the 189 patients with low fT3 levels, there was a significant disparity between survivors and deceased patients (*p* < 0.001). Of these, 32.3% had low fT3 levels both before and during ICU admission (prolonged ESS), with 26.1% survivors and 58.3% deceased (*p* < 0.001). NIV failure was significantly more common in the ESS group (44.4%) than in the non-ESS group (26.4%; *p* < 0.001). ***Conclusion:*** Our study demonstrates that ESS, mainly characterized by low fT3 levels, was significantly associated with increased mortality rates in ICU patients and a higher failure rate of NIV.

## 1. Introduction

Euthyroid sick syndrome (ESS), or non-thyroidal illness syndrome (NTIS), is characterized by abnormal thyroid function test results in the absence of underlying thyroid disease. It is commonly seen in critically ill patients and is linked to changes in the hypothalamic–pituitary–thyroid axis [1].

The syndrome typically manifests as low serum free triiodothyronine (fT3), normal or low free thyroxine (fT4), and normal or low thyroid-stimulating hormone (TSH) levels [2]. This condition’s most prominent and defining characteristic is a marked reduction in T3 hormone levels [3]. Consequently, this disorder is often called low-fT3 syndrome [4]. Initially, ESS was considered an adaptive mechanism to reduce energy expenditure and protein catabolism in response to critical illness [1]. However, subsequent studies have shown that ESS may also serve as a significant indicator of prognosis and mortality [5,6,7].

Hypothyroidism itself contributes to ventilator-dependent respiratory failure via several mechanisms, including impaired responses to hypercapnia and hypoxia, respiratory muscle dysfunction, and pleural effusions [8]. Muscle biopsies in hypothyroidism have shown Type II fiber atrophy and up to 50% loss of muscle mass [9]. These findings are thought to be associated with increased membrane permeability and decreased adenosine triphosphate production, reflected by elevated creatine kinase levels [10]. On the other hand, Euthyroid sick syndrome (ESS) is characterized by low fT3 levels, and although this may potentially affect the performance of respiratory muscles, its impact on respiratory failure has not been clearly demonstrated. Comprehensive and controlled studies are needed to clarify this matter. Nevertheless, low fT3 may compromise non-invasive ventilation (NIV) efficacy, increasing treatment failure risk.

Risk factors for NIV failure in critically ill patients with ESS include the severity and duration of the underlying illness, as assessed by scoring systems such as the Acute Physiology and Chronic Health Evaluation II (APACHE II) or Sequential Organ Failure Assessment (SOFA), with higher scores indicating greater risk [11]. Prolonged ESS exacerbates respiratory dysfunction and is often accompanied by multiple organ dysfunction syndromes affecting the cardiovascular, renal, and hepatic systems, further complicating respiratory outcomes [12].

Malnutrition, commonly seen in critically ill patients, weakens respiratory muscle function and overall resilience. At the same time, advanced age and comorbidities such as Chronic Obstructive Pulmonary Disease (COPD), heart failure, and neuromuscular disorders also elevate the risk of NIV failure.

Non-invasive mechanical ventilation is widely utilized in acute respiratory failure management. However, NIV failure is a persistent challenge in patients with complex conditions like ESS [13]. While several studies have examined the link between ESS and outcomes in invasive mechanical ventilation [14], data specific to NIV are lacking. To our knowledge, this is the first study to explore the association between ESS, NIV failure, and in-hospital mortality in critically ill patients.

## 2. Materials and Methods

### 2.1. Setting and Study Design

This study was conducted in Konya City Hospital, which comprises three internal ICUs, with a total of 45 beds. Patients who were monitored in the ICU from 1 April 2021 to 30 April 2024, received non-invasive mechanical ventilation therapy, and had serum fT3, fT4, and TSH function tests were included in the study. The study and data analysis were conducted according to the Helsinki Declaration and approved by the Ethics Committee of KTO Karatay University (approval number 2023/019-64223). Because this was a retrospective study, written informed consent was not obtained from the patients.

A total of 2081 patients were admitted to the ICU within the specified dates above. A flow chart of the patients included in this study is shown in Figure 1. Patients were excluded from the study if they had an ICU stay of less than 24 h; NIV usage of less than 24 h; pregnancy, thyroid, or hypothalamic-pituitary disease (disorders originating from the thyroid gland itself or the hypothalamic–pituitary axis that directly affect thyroid hormone production, such as autoimmune thyroiditis, pituitary adenomas, or hypothalamic lesions); use of iodinated contrast material within the previous 8 weeks; or a history of thyroid medication use. Additionally, to exclude primary hypothyroidism, patients with high serum TSH levels and low fT3 and fT4 levels, as well as clinically overt hypothyroid symptoms, were excluded. To exclude central (secondary) hypothyroidism, patients with low or inappropriately normal TSH levels along with low fT3 and fT4 levels were not included in the study. Additionally, patients with thyrotoxicosis (hyperthyroidism), characterized by suppressed TSH levels and elevated fT3 and fT4 levels, were also excluded from the study. After applying the exclusion criteria, 386 patients were included in the study.

The patients were categorized into two groups: those with ESS and those without (non-ESS). The normal ranges for thyroid hormone levels in our hospital laboratory are fT3, 2.00–4.40 ng/L; fT4, 9.3–17 ng/L; and TSH, 0.27–4.2 mU/L. Patients with fT3 levels < 2.00 ng/L were assigned to the ESS group, while those with fT3 levels ≥ 2.00 ng/L were assigned to the non-ESS group. Additionally, among the 189 patients in the ESS group, 80 were found to have concurrent low fT4 levels. Furthermore, in 61 patients in the ESS group, low fT3 levels were detected in laboratory tests conducted retrospectively within the previous 6 months before ICU admission, who were defined as the prolonged ESS group and evaluated separately (Figure 1).

Non-invasive ventilation treatment was initiated in patients presenting with acute respiratory failure characterized by respiratory acidosis (PaCO_2_ ≥ 45 mmHg and/or pH ≤ 7.2) and hypoxia (PaO_2_ ≤ 60 mmHg) in the absence of any contraindications [15].

The intensive care specialist, who continuously monitored the patients, defined ventilatory failure as an increase in PaCO_2_ levels in arterial blood gas analysis, an increase in respiratory rate, respiratory distress, work of breathing, deepening of respiratory acidosis, a decrease in tidal volume, arterial oxygenation, and deterioration in consciousness [15].

### 2.2. Data Collection

The laboratory and medical records of the patients were retrieved from the electronic database and hospital files. Comorbidities, indications for intensive care unit hospitalization, arterial blood gas values (pH, PaCO_2_, HCO_3_, and PaO_2_), thyroid function tests (fT3, fT4, and TSH), APACHE II, Glasgow Coma Scale (GCS), and SOFA scores were recorded within 24 h of admission. Furthermore, the use of NIV and continuous oxygen therapy at the patient’s residence were documented.

To enhance data reliability and minimize the observational bias inherent in retrospective study designs, clinical and laboratory data were independently extracted and cross-verified by two experienced intensive care specialists. These physicians reviewed electronic medical records, nursing documentation, and laboratory systems in parallel to ensure the completeness, consistency, and accuracy of all variables included in the analysis. Any discrepancies identified during data abstraction were resolved through consensus, thereby improving the internal validity of the dataset used for statistical evaluation.

### 2.3. Primary Secondary Outcomes

The principal object of this study was to assess the effect of ESS on NIV failure in patients undergoing NIV treatment for respiratory distress in the ICU. Furthermore, the impact of ESS on ICU length of stay and ICU mortality in the ESS group was evaluated. This study also investigated the effect of prolonged low fT3 levels before admission to the ICU on mortality and NIV failure.

### 2.4. Statistical Analyses

All data were analyzed using IBM SPSS software (version 26.0, Armonk, NY, USA). The Kolmogorov–Smirnov test was used to assess the normality of data distribution. Quantitative variables with a normal distribution are presented as mean ± standard deviation (SD), while those not normally distributed are reported as medians with interquartile ranges (IQR). Qualitative variables are expressed as frequencies and percentages.

For comparisons of continuous variables, Student’s *t*-test was applied to normally distributed data, and the Mann–Whitney U test was used for non-normally distributed data. Categorical variables were compared using the chi-square test or Fisher’s exact test, as appropriate. Group comparisons for clinical outcomes (NIV success/failure and ICU discharge/death) were performed using a chi-square test.

To identify factors associated with NIV failure and ICU mortality, univariate and multivariable logistic regression analyses were conducted. Variables with a *p*-value < 0.1 in the univariate analysis were included in the multivariable logistic regression model to account for a broader range of potentially relevant factors in the exploratory analysis. Potential confounders, including age, sex, APACHE II score, comorbidities (e.g., diabetes, cardiovascular disease), lactate levels, and length of stay in the ICU, were adjusted for in the multivariable analysis. To avoid multicollinearity, variables with high correlation (Pearson’s r > 0.7), such as low fT3 group, fT3, fT4 levels, and NIV time, were excluded from the model. The backward stepwise method was used to identify independent predictors. The goodness-of-fit of the multivariable model was assessed using the Hosmer–Lemeshow test, and model performance was evaluated with Nagelkerke R^2^.

Receiver Operating Characteristic (ROC) curve analyses were performed to evaluate the predictive ability of the thyroid function tests (e.g., fT3 levels) for NIV failure and ICU mortality. Area Under the Curve (AUC) values, along with 95% confidence intervals (CI), were calculated to assess the discriminatory performance. A *p*-value < 0.05 was considered statistically significant for all analyses.

Due to the retrospective nature of the study, potential limitations, such as missing data or unmeasured confounders (e.g., pre-ICU nutritional status), may have affected the analysis. These are further discussed in the Limitations section, along with recommendations for prospective studies to address these issues using standardized thyroid hormone measurements and improved confounder control.

## 3. Results

### 3.1. Patient Demographics and Clinical Characteristics

The study included 386 patients, comprising 332 survivors and 54 deceased individuals. ESS was observed in 49.0% of the total patient population, with a significantly higher prevalence among deceased patients (66.7%) than among survivors (46.1%) (Table 1). In the group with low fT3 levels before ICU admission (prolonged ESS group), the percentage of patients with normal fT4 levels was 27.9%, and the rate of patients with concurrent low fT4 levels was 72.1%. In contrast, 28.1% of the patients with normal fT3 levels before ICU admission had low fT4 levels (*p* < 0.001). While 84.4% of patients with normal fT4 levels were in the newly diagnosed ESS group, only 45.0% of patients with low fT4 levels fell into this category. Conversely, 15.6% of patients with normal fT4 levels were in the prolonged ESS group, compared to 55.0% in the low fT4 group (*p* < 0.001).

### 3.2. Clinical Scores and Laboratory Findings

Significant differences were noted for various clinical scores and laboratory findings between the deceased patients and survivors (Table 1). Deceased patients had notably higher SOFA and APACHE II scores (*p* < 0.001) than survivors. Additionally, the deceased patients exhibited significantly lower fT3 levels (*p* < 0.002) and higher fT4 levels (*p* = 0.011). Elevated lactate levels (*p* < 0.001) and longer durations of NIV (*p* < 0.001) were also more common among deceased patients. Moreover, elevated CRP levels were significantly associated with mortality (*p* = 0.002). In the ESS group, higher HCO_3_ levels (*p* = 0.003) and longer durations of NIV (*p* < 0.001) were observed than in the non-ESS group. However, there were no significant differences in Ph, PCO_2_, PO_2_, lactate, or CRP levels (*p* > 0.05) (Table 2).

### 3.3. Mortality and NIV Failure

Among the 189 patients with low fT3 levels, there was a significant disparity between survivors and deceased patients (*p* < 0.001). Of these, 67.7% had low fT3 levels within the normal range before ICU admission, with 73.9% being survivors and 41.7% being deceased. In contrast, 32.3% had low fT3 levels both before and during ICU admission, with 26.1% survivors and 58.3% deceased. An analysis of the ESS subgroups also revealed significant differences (*p* < 0.001), with 57.7% having normal fT4 and 42.3% having low fT4. Among the survivors, 64.1% had normal fT4 levels, while 30.6% had normal fT4 levels. In the low-fT4 subgroup, 35.9% were survivors compared to 69.4% deceased patients. A significant difference was observed in the ICU mortality rates, which were higher in the ESS group (19.0%) than in the non-ESS group (9.1%; *p* = 0.005). Similarly, NIV failure was significantly more common in the ESS group (44.4%) than in the non-ESS group (26.4%; *p* < 0.001) (Table 2). ESS was present in 42.0% of the group without NIV failure compared with 61.8% in the NIV failure group (*p* < 0.001) (Table 3).

Among the 189 patients with low fT3 levels, there were significant differences between the non-NIV failure and NIV failure groups in the subgroup analysis (*p* = 0.002). While 77.1% of the non-NIV failure group had fT3 levels in the normal range before ICU admission, only 56.0% of the NIV failure group fell into this category. Furthermore, 22.9% of the non-NIV failure group had low fT3 levels both before and during ICU admission, compared to 44.0% in the NIV failure group (Table 3). A distribution difference was observed in the ESS subgroups with normal fT4 levels (*p* = 0.107); 62.9% of the non-NIV failure group had normal fT4 levels compared to 51.2% in the NIV failure group. The low-fT4 subgroup comprised 37.1% of the non-NIV failure group and 48.8% of the NIV failure group (Table 3). The ICU mortality rate was significantly higher in the NIV failure group (36.0%) than in the non-NIV failure group (2.0%; *p* < 0.001) (Table 3).

Table 4 ESS compares subgroups with (prolonged ESS groups) and without (newly diagnosed ESS groups) low serum fT3 levels before ICU admission. Patients with low fT3 levels had a significantly higher ICU mortality rate (34.4%) than those without low fT3 levels (11.7%; *p* < 0.001). NIV failure in the ICU was also more common in patients with low fT3 levels (60.7%) than in those without low fT3 levels (36.7; *p* = 0.002) (Table 4). The ICU mortality rate was significantly higher in patients with low fT4 levels (31.3%) than in those with normal fT4 levels (10.1%; *p* < 0.001). Additionally, NIV failure in the ICU was more frequent in the low-fT4 groups (51.2%) than in the normal-fT4 group (39.4), although the difference was not statistically significant (*p* = 0.107). However, duration of NIV was longer in the low-fT4 group (*p* < 0.001) (Table 5). Table 6 presents multivariable analyses identifying the factors independently associated with ICU mortality and NIV failure. Having ESS was found to be an independent factor related to ICU mortality regression models (Table 6). The ROC curve analyses showed the best cut-off values for thyroid function test levels for predicting ICU mortality and NIV failure (Table 7). For predicting ICU mortality, the area under the curve (AUC) for fT3 was 0.631 (*p* = 0.002), and for predicting NIV failure, the AUC for fT3 was 0.630 (*p* < 0.001).

### 3.4. Length of ICU Stay

The length of ICU stay was significantly longer for deceased patients (Table 1). Similarly, patients in the ESS group had a notably more extended ICU stay, with a median of 8 days (IQR 4–18) compared with 6 days (IQR 3–15) in the non-ESS group (*p* < 0.001) (Table 2) (Figure 2). Furthermore, the ICU stay duration was significantly longer for patients with NIV failure, with a median of 10 days (IQR 4–18) compared to 6 days (IQR 3–15) for those without NIV failure (*p* < 0.001) (Table 3). Patients with low fT3 levels also experienced longer ICU stays, with a median of 10 days (IQR 4–18) compared to 7.5 days (IQR 4–18) for those with normal fT3 levels (*p* < 0.001) (Table 4). Additionally, patients with low fT4 levels had a significantly longer ICU stay, with a median of 10 days (IQR 4–18) compared to 7 days (IQR 4–12) for those with normal fT4 levels (*p* < 0.001) (Table 5).

## 4. Discussion

To our knowledge, this study represents a pioneering effort to comprehensively assess the impact of ESS on NIV failure and ICU mortality. Our findings revealed a statistically significant correlation between ESS and ICU mortality and NIV failure. Furthermore, an analysis of patients with low fT3 levels before ICU admission demonstrated that prolonged ESS was an important factor for NIV failure. This underscores the crucial role of thyroid function in managing critically ill patients in the ICU setting.

Although ESS was initially perceived as an adaptive mechanism wherein the body reduces thyroid hormone production to conserve energy under stress, as demonstrated in previous studies, it can lead to muscle wasting in critically ill patients and result in adverse outcomes such as increased mortality [16] and exacerbation of respiratory failure, due to its respiratory [17] and cardiac effects [18]. NIV is an important treatment modality for acute respiratory failure in the ICU. NIV failure may necessitate the transition of critically ill patients to invasive mechanical ventilation, thereby exposing them to numerous potential complications. Consequently, our study underscores the significance of early diagnosis and monitoring of thyroid hormone disorders in the ICU setting, as such measures can potentially improve patient outcomes, including mortality rates and ICU length of stay.

There have been many studies on ESS in the literature. These studies have consistently demonstrated that ESS is associated with poor prognostic outcomes for various organ failures, including cardiovascular diseases [19], acute neurological events [20], and renal failure [21]. Moreover, low serum fT3 levels have been linked to detrimental effects such as cognitive function impairment in intensive care patients [22], respiratory muscle weakness, and decreased lung compliance [8,9]. Such findings underscore the complex and multifaceted impact of thyroid hormone disturbances in critically ill patients and highlight the importance of the early detection and management of ESS to improve clinical outcomes.

In the present study, the prevalence of ESS was 49%. This rate was 54.38% in a study by Wang et al. [23] and 38.7% in another study [24]. We attribute these variations to the severity of patients’ illnesses and the distinctions between medical and surgical patient groups. This indicates that ESS becomes more prevalent over the long term, particularly as the incidence of chronic diseases and their adverse effects increase. Additionally, our findings revealed that ESS was significantly higher in deceased patients than in survivors in all-patient groups. This aligns with previous studies suggesting EES is associated with poorer outcomes in critically patients [14,15,16,17,18,19,20,21,22,23,24,25].

Previous studies have demonstrated a significant association between low fT3 levels and ICU mortality [26]. Our study corroborates these findings, as we observed a high mortality rate in the ESS group. Furthermore, our results indicate that fT3 levels were significantly associated with ICU mortality scores, including SOFA and APACHE II [26].

Most previous studies have shown that ESS is largely associated with disease severity [5,6,7,8,9,10,11,12,13,14,15,16,17,18,19,20,21,22,23,24,25,26]. In our study, lactate levels, APACHE II, and SOFA scores observed in the ESS group were statistically significant in terms of NIV failure in a multivariable logistic regression analysis compared to the non-ESS group (*p* < 0.001). However, CRP levels, which could also be related to disease severity, did not show a significant difference between the two groups in our study (*p* = 0.106). Nonetheless, the median CRP level was higher in the ESS group, with a *p*-value close to the borderline significance level. Fastiggi et al. demonstrated in their study that ESS had a high correlation with CRP and various inflammatory markers but was not correlated with disease duration [27]. The lack of high CRP levels in our ESS patients may be related to the disease stage and the heterogeneity of patients in the ESS group, as CRP is an early indicator of the inflammatory response. This unexpected finding highlights the necessity for further research to understand the pathophysiology of ESS and its effects on the inflammatory response.

Unlike other studies, a key aspect of our research was the retrospective analysis of patient data within six months prior to ICU admission, which we categorized as the prolonged ESS group. We observed a stronger statistical significance for mortality in patients with prolonged low serum fT3 levels. This finding suggests that the longer the ESS persists, the stronger the association with poor prognosis. Consequently, it is imperative to meticulously assess patients’ thyroid hormone levels upon ICU admission and implement appropriate nutritional management. In addition, specific treatment strategies should be tailored to address sarcopenia and frailty in these patients.

In our study, we found that low fT3 levels were significantly associated with NIV failure in univariate analysis (*p* = 0.002), but in multivariable analysis, low fT3 levels did not emerge as an independent prognostic factor. This can be explained by the influence of other strong prognostic factors such as high APACHE II and SOFA scores. Possible reasons to consider are that other critical factors in the multivariable analysis might have masked the effect of low fT3, or our sample size may not have been sufficiently powered. Additionally, due to the complex pathophysiology of ESS, its contribution to NIV failure might not be direct and evident. It is also important to consider that in a cumulative risk model, risk factors might not show a significant effect individually, but may contribute to the overall risk and to NIV failure through synergistic and additive effects. A retrospective observational cohort study by Krug et al. identified the duration of mechanical ventilation, acute kidney injury, sepsis, and acute liver failure as independent predictors of mortality in ESS patients. However, ESS itself was not reported as an independent risk factor for increased ICU mortality in a multivariable logistic regression analysis [28]. Similarly, in our study, low fT3 levels were associated with NIV failure in the univariate analysis, but this relationship was not found to be an independent factor in the multivariable analysis. These findings suggest that ESS alone is not sufficient to determine NIV failure and that NIV failure is a multifactorial event. Larger-scale, prospective, and multicenter studies are needed in the future to better understand the impact of ESS on NIV efficacy and overall patient outcomes.

Our study determined the optimal cut-off value for serum fT3 as ≤1.33 based on the ROC analysis. This threshold exhibited moderate sensitivity and high specificity, suggesting its utility as a valuable prognostic marker for predicting ICU mortality and NIV success.

In the analysis of all-cause mortality among survivors, high fT4 levels were also associated with increased mortality, albeit with a lower specificity. This could have been due to disruptions in the conversion of fT4 to fT3, possibly caused by the use of drugs such as glucocorticoids, which inhibit 5′ deiodinase, or acute inhibition of the enzyme iodothyronine deiodinase in the early phases, or an increase in thyroxine-binding globulin levels [29]. In our analysis of the ESS subgroups, patients with low fT3 levels before ICU admission exhibited significantly lower fT4 levels, which strongly correlated with higher ICU mortality rates, prolonged NIV duration, and extended ICU stays. These findings suggest that initial fT4 levels may be in normal ranges, but their subsequent decline is linked to worse clinical outcomes.

Our study also underscored the critical role of ESS in predicting NIV failure. The rate of NIV failure was significantly higher in patients with ESS than in those without ESS, with low fT3 levels showing a strong association with NIV failure. A cut-off value of fT3 <1.77 demonstrated moderate sensitivity and specificity. Additionally, NIV failure rates were notably higher in the prolonged ESS group.

As a secondary outcome of a recent study investigating the relationship between ESS and prognosis in patients with sepsis, ESS and NIV failure were evaluated, but no association was found [30]. By contrast, our study identified ESS as an independent and robust risk factor for NIV failure. This discrepancy may stem from the primary outcome focus of the other study, which did not focus on NIV failure. We believe future studies examining the relationship between ESS and NIV failure will corroborate our findings.

Furthermore, patients with low serum fT4 levels experienced NIV failure more frequently, although this was not statistically significant (*p* = 0.107), indicating that fT3 may be a more reliable marker. This finding might have been influenced by the retrospective nature and relatively small sample size of the present study. Previous studies have shown that persistently low fT4 levels may be associated with prolonged mechanical ventilation and increased mortality [12,13,14]. However, in our study, low serum fT4 levels were associated with longer durations of NIV use.

In our study, ESS was associated with longer ICU stays. Similarly, in the subgroup analyses, both groups with prolonged ESS had longer ICU stays than those without ESS, and patients with low fT4 levels stayed longer in the ICU than those without ESS.

The findings of this study underscore the importance of regular monitoring of thyroid function in patients in the ICU. The early detection of ESS and adjustment of clinical management strategies could potentially improve patient outcomes. The significant association between ESS, particularly low fT3 levels, and critical outcomes such as ICU mortality and NIV failure suggests that thyroid hormone levels should be a key consideration in the routine assessment of critically ill patients. The relationship between thyroid hormone replacement and improvements in mortality and clinical outcomes has been analyzed in patients with ESS-associated heart failure [31] and septic shock [32], with findings indicating its effectiveness. Given the observed association between NIV failure and low fT3 levels, prospective studies are warranted to investigate the potential benefits of thyroid hormone replacement therapy in improving the clinical outcomes of patients undergoing NIV.

This study had several limitations. First, being retrospective and including a single center may have introduced a selection bias and limit the generalizability of the findings. Additionally, reliance on previously recorded data in a retrospective study may be prone to information bias, as not all relevant clinical variables may have been consistently documented. The relatively small sample size, particularly among patients with ESS and those experiencing NIV failure, may have limited the statistical power of our study. This point could have impacted the robustness of our findings and their generalizability to larger and more diverse patient populations. Another limitation of this study was the lack of a longitudinal assessment of thyroid function. While the study primarily focused on thyroid function test results obtained at the time of the ICU admission, it did not systematically analyze changes in thyroid hormone levels during the ICU stay. Longitudinal data could provide valuable insights into the dynamic nature of thyroid function in critically ill patients and its impact on clinical outcomes. Additionally, this study did not account for potential confounding factors such as the severity of underlying comorbidities, use of specific medications, or other interventions that might affect thyroid function and patient outcomes. These factors may have contributed to the observed associations and should be considered in future studies. Finally, although this study assessed the association between ESS, NIV failure, and ICU mortality, causality was not established, due to cross-sectional and retrospective design. Prospective studies with larger sample sizes and multicenter designs must confirm these findings and further investigate the mechanisms underlying this association. This retrospective analysis provides a basis for prospective studies aimed at evaluating whether targeted interventions for ESS can improve clinical outcomes in the ICU.

Additionally, the observed differences in the prevalence of ESS and its associations with clinical outcomes, when compared to prior studies, may be attributed to several methodological and population-related factors. These include heterogeneity in ICU patient populations across studies (e.g., medical vs. surgical ICU cohorts), variability in the operational definitions of ESS (such as differing fT3 cut-off thresholds), and inconsistencies in the timing of thyroid hormone measurements relative to the onset of critical illness. Such variations may have influenced the interpretation and comparability of the results. Therefore, the findings of the current study should be contextualized within its specific design and patient characteristics, and future prospective multicenter studies with standardized ESS criteria are needed to validate and generalize these observations. Furthermore, assessing pre-ICU functional and nutritional status could better contextualize thyroid function trends. Interventional studies with longitudinal follow-up to examine the impact of thyroid hormone modulation on NIV outcomes and mortality are necessary to confirm causal relationships.

Despite these limitations, our study has several strengths. First, to the best of our knowledge, this is the first study to specifically target ESS and NIV failures in this population. Second, we conducted a retrospective analysis of six-month patient records before ICU admission, including patients with prolonged low fT3 levels. This makes our study a pioneering effort in examining the impact of long-term ESS in the ICU setting.

## 5. Conclusions

Our study demonstrated a significant association between ESS and mortality rates in ICU patients, particularly marked by low fT3 levels, increased mortality rates, and a higher failure rate of non-invasive ventilation. These results underscore the prognostic importance of thyroid function tests in critically ill patients and suggest that addressing ESS could enhance patient outcomes. Future research should investigate this association’s mechanisms and assess potential therapeutic interventions to correct thyroid dysfunction in a critical care environment.

Taken together, these findings underscore the importance of designing prospective cohort studies with standardized protocols for thyroid hormone assessment in critically ill patients. Such studies are essential for establishing a causal relationship between ESS and adverse clinical outcomes, and to explore whether timely identification and targeted therapeutic interventions such as thyroid hormone replacement may improve NIV success rates and reduce ICU mortality. Future multicenter investigations with larger and more diverse populations are warranted to validate these preliminary findings and guide evidence-based clinical decision-making regarding thyroid dysfunction in the ICU setting.

## Figures and Tables

**Figure 1 medicina-61-00927-f001:**
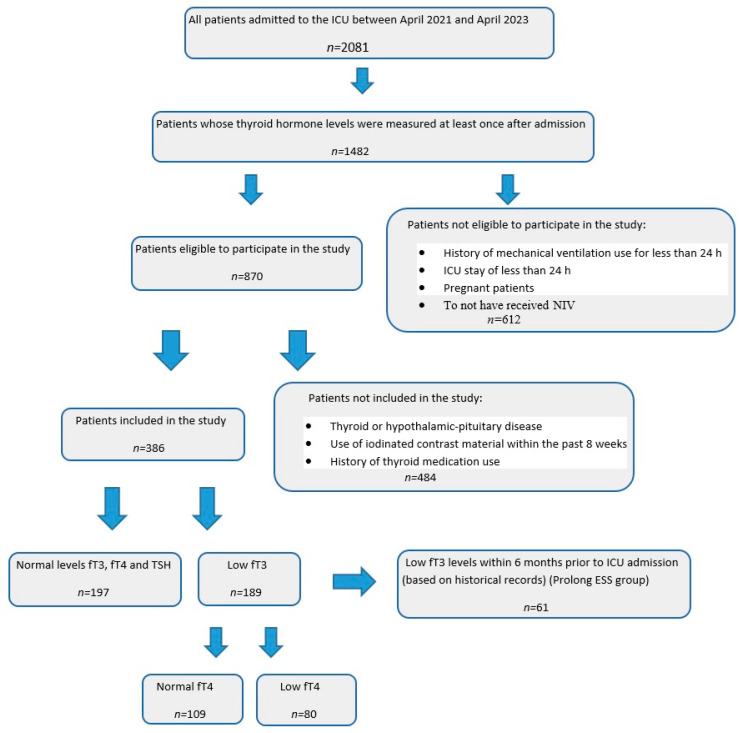
Flow chart of the study patient enrollment.

**Figure 2 medicina-61-00927-f002:**
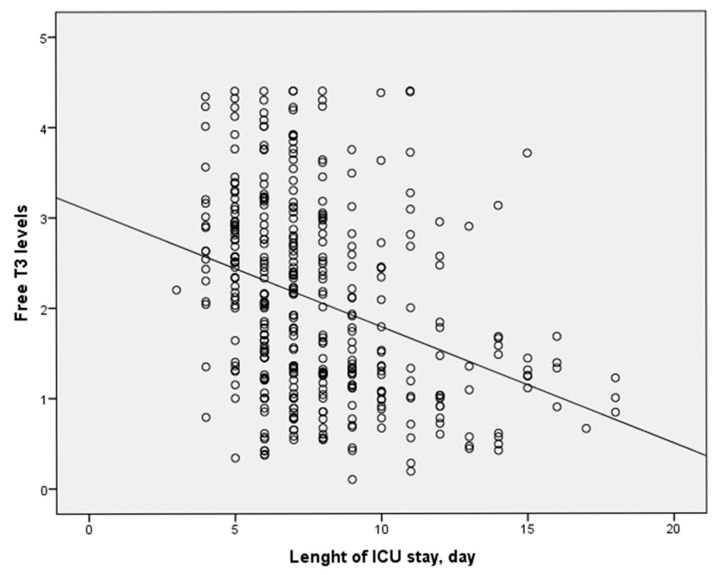
This figure shows there was a negative, moderate, and statistically significant correlation between fT3 and LOS-ICU levels (rho = −0.360 and *p* < 0.001).

**Table 1 medicina-61-00927-t001:** Comparison of study variables between deceased and surviving patients.

Parameters	Total Patients *n* = 386	Survived *n* = 332	Deceased *n* = 54	*p*-Value
Age, years	65 (36-89)	65 (43-89)	66 (36-79)	0.212
Gender, male	205 (53.1)	152 (45.8)	29 (53.7)	0.279
Patient’s groups				0.005
Non-euthyroid sick syndrome	197 (51.0)	179 (53.9)	18 (33.3)	
Euthyroid sick syndrome (ESS)	189 (49.0)	153 (46.1)	36 (66.7)	
Low fT3 (total *n* = 189)				<0.001
Low fT3 (but in the normal range before ICU admission) (Newly Diagnosed ESS groups)	128 (67.7)	113 (73.9)	15 (41.7)	
Low fT3 (Prolonged ESS groups)	61 (32.3)	40 (26.1)	21 (58.3)	
ESS subgroups (total *n* = 189)				<0.001
with normal fT4	109 (57.7)	98 (64.1)	11 (30.6)	
with low fT4	80 (42.3)	55 (35.9)	25 (69.4)	
Reasons for ICU admission				
Cardiopulmonary edema	112 (29.0)	100 (30.1)	12 (22.2)	0.236
Chronic obstructive pulmonary disease (COPD)	122 (31.6)	99 (29.8)	23 (42.6)	0.061
Pneumonia	120 (31.1)	106 (31.9)	14 (25.9)	0.377
Post-extubation respiratory failure	20 (5.2)	18 (5.4)	2 (3.7)	1.000
Obesity hypoventilation syndrome	8 (2.1)	7 (2.1)	1 (1.9)	1.000
Postoperative respiratory failure	4 (1.0)	4 (1.2)	2 (3.7)	0.199
Comorbidities				
Diabetes mellitus	218 (56.5)	194 (58.4)	24 (44.4)	0.054
Coronary artery disease	185 (47.9)	162 (48.8)	23 (42.6)	0.397
Hypertension	174 (45.1)	151 (45.5)	23 (42.6)	0.692
Chronic kidney disease	28 (7.3)	25 (7.5)	3 (5.6)	0.781
Other clinical variables				
Home NIV	58 (15.0)	44 (13.3)	14 (25.9)	0.016
Home long-term oxygen therapy	103 (26.7)	86 (25.9)	17 (31.5)	0.390
NIV failure in the ICU	136 (35.2)	87 (26.2)	49 (90.7)	<0.001
CCI, score >3 points	175 (45.1)	156 (47.0)	19 (35.2)	0.106
SOFA score	9 (0–17)	8 (0–17)	12 (3–15)	<0.001
APACHE II	18 (15–29)	18 (15–29)	24.5 (15–29)	<0.001
Glasgow Coma Scale	15 (12–15)	15 (12–15)	15 (13–15)	0.920
NIV time day	5 (2–14)	5 (2–13)	7 (4–14)	<0.001
Length of ICU stay, day	7 (3-18)	7 (3–18)	11 (6–18)	<0.001
Laboratory findings				
pH	7.19 (7.05–7.33)	7.19 (7.05–7.33)	7.21 (7.05–7.33)	0.373
pCO_2_	60 (47–75)	60 (48–75)	58.5 (47–73)	0.162
HCO_3_	33 (26–43)	33 (26–43)	31.5 (27–43)	0.090
PO_2_	74 (53–88)	74 (53–88)	72 (61–88)	0.254
Lactate	1.8 (1–3.2)	1.7 (1–2.7)	2.3 (1–3.2)	<0.001
TSH	1.69 (0.18–4.2)	1.7 (0.21–4.2)	1.64 (0.18–4.12)	0.670
fT3	2.02 (0.1–4.4)	2.11 (0.1–4.4)	1.3 (0.19–4.4)	0.002
fT4	12.41 (9.2–17)	12.38 (9.2–17)	13.32 (9.3–17)	0.011
CRP	103.63 (1.1–198.83)	99.11 (1.1–198.83)	123.86 (18.6–193.97)	0.002

ESS, Euthyroid sick syndrome; COPD, Chronic obstructive pulmonary disease; NIV, Non-invasive mechanical ventilation; ICU, Intensive care unit; CCI, Charlson’s comorbidity index; SOFA, Sequential Organ Failure Assessment; APACHE II, Acute Physiology and Chronic Health Evaluation; TSH, thyroid-stimulating hormone; CRP, C-Reactive Protein. Data are reported as mean ± SD or percentage n (%) or median (min-max). Continuous and categorical variables were compared between groups with the Mann–Whitney U test or Pearson’s chi-squared test, respectively. *p* < 0.05 was considered statistically significant in all analyses.

**Table 2 medicina-61-00927-t002:** Comparison of study variables between patients with and without ESS.

Parameters	Non-ESS Group *n* = 197	ESS Group *n* = 189	*p*-Value
Gender, male	110 (55.8)	95 (50.3)	0.273
Age, years	65 (43–89)	65 (36-86)	0.780
Reasons for ICU admission			
Cardiopulmonary edema	58 (29.4)	54 (28.6)	0.851
Chronic obstructive pulmonary disease (COPD)	60 (30.5)	62 (32.8)	0.620
Pneumonia	61 (31.0)	59 (31.2)	0.957
Post-extubation respiratory failure	9 (4.6)	11 (5.8)	0.579
Obesity hypoventilation syndrome	6 (3.0)	2 (1.1)	0.285
Postoperative respiratory failure	3 (1.5)	3 (1.6)	1.000
Comorbidities			
Diabetes mellitus	115 (58.4)	103 (54.5)	0.442
Coronary artery disease	99 (50.3)	86 (45.5)	0.350
Hypertension	91 (46.2)	83 (43.9)	0.653
Chronic kidney disease	17 (8.6)	11 (5.8)	0.287
Other clinical variables			
Home NIV	33 (16.8)	25 (13.2)	0.333
Home long-term oxygen therapy	58 (29.4)	45 (23.8)	0.211
ICU mortality rate	18 (9.1)	36 (19.0)	0.005
NIV failure in the ICU	52 (26.4)	84 (44.4)	<0.001
CCI, score > 3 points	90 (45.7)	85 (45.0)	0.888
SOFA score	8 (1–17)	10 (0–17)	0.089
APACHE II	17 (15–29)	18 (15–29)	<0.001
Glasgow Coma Scale	15 (12–15)	15 (12–15)	0.257
NIV time day	4 (2–8)	6 (3–14)	<0.001
Length of ICU stay, day	6 (3–15)	8 (4–18)	<0.001
Laboratory findings			
pH	7.19 (7.05–7.33)	7.19 (7.05–7.33)	0.927
pCO_2_	60 (48–73)	60 (47–75)	0.717
HCO_3_	32 (26–43)	34 (26–43)	0.003
PO_2_	74 (53–88)	74 (53–88)	0.718
Lactate	1.7 (1–2.8)	1.8 (1–3.2)	0.200
TSH	1.54 (0.27–4.2)	1.85 (0.18–4.12)	0.058
fT3	2.87 (2–4.4)	1.2 (0.1–1.93)	<0.001
fT4	12.4 (9.3–17)	12.49 (9.2–17)	0.429
CRP	99.7 (1.5–198.83)	105.81 (1.1–198.26)	0.106

ESS, Euthyroid sick syndrome; COPD, Chronic obstructive pulmonary disease; NIV, Non-invasive mechanical ventilation; ICU, Intensive care unit; CCI, Charlson’s comorbidity index; SOFA, Sequential Organ Failure Assessment; APACHE II, Acute Physiology and Chronic Health Evaluation; TSH, thyroid stimulating hormone; CRP, C-Reactive Protein. Data are reported as mean ± SD or percentage n (%) or the median (min–max). Continuous and categorical variables were compared between groups with the Mann–Whitney U test or Pearson’s chi-squared test, respectively. *p* < 0.05 was considered statistically significant in all analyses.

**Table 3 medicina-61-00927-t003:** Comparison of study variables between patients with and without non-invasive ventilation failure.

Parameters	Patients Without NIV Failure *n* = 250	Patients With NIV Failure *n* = 136	*p*-Value
Gender, male	131 (52.4)	7 (54.4)	0.705
Age, years	65 (45–89)	65 (36–86)	0.814
Patient’s groups			<0.001
Non-euthyroid sick syndrome	145 (58.0)	52 (38.2)	
Euthyroid sick syndrome (ESS)	105 (42.0)	84 (61.8)	
Low fT3 (total *n* = 189)			0.002
Low fT3 (but in normal range before ICU admission)	81 (77.1)	47 (56.0)	
Low fT3 (Prolonged ESS groups)	24 (22.9)	37 (44.0)	
ESS subgroups (total *n* = 189)			0.107
with normal fT4	66 (62.9)	43 (51.2)	
with low fT4	39 (37.1)	41 (48.8)	
Reasons for ICU admission			
Cardiopulmonary edema	79 (31.6)	33 (24.3)	0.129
Chronic obstructive pulmonary disease (COPD)	78 (31.2)	44 (32.4)	0.816
Pneumonia	73 (29.2)	47 (34.6)	0.277
Post-extubation respiratory failure	13 (5.2)	7 (5.1)	0.982
Obesity hypoventilation syndrome	5 (2.0)	3 (2.2)	0.892
Postoperative respiratory failure	3 (1.2)	3 (2.2)	0.445
Comorbidities			
Diabetes mellitus	149 (59.6)	69 (50.7)	0.093
Coronary artery disease	123 (49.2)	62 (45.6)	0.497
Hypertension	123 (49.2)	51 (37.5)	0.027
Chronic kidney disease	19 (7.6)	9 (6.6)	0.722
Other clinical variables			
Home NIV	38 (15.2)	20 (14.7)	0.897
Home long-term oxygen therapy	67 (26.8)	36 (26.5)	0.944
ICU mortality rate	5 (2.0)	49 (36.0)	<0.001
CCI, score > 3 points	127 (50.8)	48 (35.3)	0.003
SOFA score	6 (0–17)	12 (4–17)	<0.001
APACHE II	17 (15–28)	22 (15–29)	<0.001
Glasgow Coma Scale	15 (12–15)	15 (13–15)	0.781
NIV time day	4 (2-9)	7 (3–14)	<0.001
Length of ICU stay, day	6 (3–15)	10 (4–18)	<0.001
Laboratory findings			
pH	7.2 (7.05–7.33)	7.19 (7.05–7.33)	0.096
pCO_2_	59 (48–73)	61 (47–75)	0.163
HCO_3_	33 (26–42)	34 (27–43)	0.269
PO_2_	74 (53–88)	73 (53–88)	0.059
Lactate	1.6 (1–2.7)	2.1 (1–3.2)	<0.001
TSH	1.67 (0.21–4.2)	1.71 (0.18–4.12)	0.705
fT3	2.21 (0.1–4.4)	1.44 (0.19–4.4)	<0.001
fT4	12.36 (9.2–17)	12.72 (9.28–17)	0.561
CRP	84.43 (1.1–198.83)	120.45 (7.46–194.13)	<0.001

ESS, Euthyroid sick syndrome; COPD, Chronic obstructive pulmonary disease; NIV, Non-invasive mechanical ventilation; ICU, Intensive care unit; CCI, Charlson’s comorbidity index; SOFA, Sequential Organ Failure Assessment; APACHE II, Acute Physiology and Chronic Health Evaluation; TSH, thyroid stimulating hormone; CRP, C-Reactive Protein. Data are reported as mean ± SD or percentage *n* (%) or the median (min–max). Continuous and categorical variables were compared between groups with the Mann–Whitney U test or Pearson’s chi-squared test, respectively. *p* < 0.05 was considered statistically significant in all analyses.

**Table 4 medicina-61-00927-t004:** Comparison of study variables between patients with and without low fT3 levels before ICU admission.

Parameters	Patients Without Low fT3 Levels Before ICU Admission *n* = 128	Patients with Low fT3 Levels Before ICU Admission (Prolonged ESS Groups) *n* = 61	*p*-Value
Gender, male	69 (53.9)	26 (42.6)	0.147
Age, years	65 (43-86)	65 (36–79)	0.622
ESS subgroups (total *n* = 189)			<0.001
with normal fT4	92 (71.9)	17 (27.9)	
with low fT4	36 (28.1)	44 (72.1)	
Reasons for ICU admission			
Cardiopulmonary edema	41 (32.0)	13 (21.3)	0.127
Chronic obstructive pulmonary disease	38 (29.7)	24 (39.3)	0.186
Pneumonia	40 (31.3)	19 (31.1)	0.989
Post-extubation respiratory failure	7 (5.5)	4 (6.6)	0.749
Obesity hypoventilation syndrome	2 (1.6)	0 (0.0)	1.000
Postoperative respiratory failure	2 (1.6)	1 (1.6)	1.000
Comorbidities			
Diabetes mellitus	81 (63.3)	22 (36.1)	<0.001
Coronary artery disease	60 (46.9)	26 (42.6)	0.583
Hypertension	62 (48.4)	21 (34.4)	0.070
Chronic kidney disease	10 (7.8)	1 (1.6)	0.090
Other clinical variables			
Home NIV	16 (12.5)	9 (14.8)	0.669
Home long-term oxygen therapy	28 (21.9)	17 (27.9)	0.366
ICU mortality rate	15 (11.7)	21 (34.4)	<0.001
NIV failure in the ICU	47 (36.7)	37 (60.7)	0.002
CCI, score > 3 points	66 (51.6)	19 (31.1)	0.008
SOFA score	8.5 (0–17)	11 (3–15)	<0.001
APACHE II	18 (15–29)	19 (15–28)	0.057
Glasgow Coma Scale	15 (12–15)	15 (12–15)	0.818
NIV time day	6 (3–13)	7 (3–14)	Table 7
Length of ICU stay, day	7.5 (4–18)	10 (4–18)	<0.001
Laboratory findings			
pH	7.19 (7.05–7.33)	7.19 (7.07–7.33)	0.870
pCO_2_	59.5 (47–73)	61 (47–75)	0.759
HCO_3_	35 (27–43)	33 (26–42)	0.080
PO_2_	74 (53–88)	74 (58–88)	0.537
Lactate	1.7 (1–2.6)	1.9 (1–3.2)	0.002
TSH	1.78 (0.18–4.11)	1.92 (0.21–4.12)	0.287
fT3	1.26 (0.1–1.93)	1.09 (0.34–1.84)	0.188
fT4	12.4 (9.2–17)	13.2 (9.28–17)	0.132
CRP	106.9 (1.1–198.26)	103.61 (2.6–186.6)	0.994

ESS, Euthyroid sick syndrome; COPD, Chronic obstructive pulmonary disease; NIV, Non-invasive mechanical ventilation; ICU, Intensive care unit; CCI, Charlson’s comorbidity index; SOFA, Sequential Organ Failure Assessment; APACHE II, The Acute Physiology and Chronic Health Evaluation; TSH, thyroid stimulating hormone; CRP, C-Reactive Protein. Data are reported as mean ± SD or percentage *n* (%) or the median (min-max). Continuous and categorical variables were compared between groups with the Mann-Whitney U test or Pearson’s chi-squared test, respectively. *p* < 0.05 was considered statistically significant in all analyses.

**Table 5 medicina-61-00927-t005:** Comparison of categorical variables between patients with normal and low fT4 levels in the ESS group.

Parameters	Patients with Normal fT4 *n* = 109	Patients with Low fT4 *n* = 80	*p*-Value
Gender, male	57 (52.3)	42 (52.5)	0.515
Low fT3 (total *n* = 189)			<0.001
Low fT3 (but in normal range before ICU admission)	92 (84.4)	36 (45.0)	
Low fT3 (Prolonged ESS groups)	17 (15.6)	44 (55.0)	
Reasons for ICU admission			
Cardiopulmonary edema	35 (32.1)	19 (23.8)	0.209
Chronic obstructive pulmonary disease (COPD)	32 (29.4)	30 (37.5)	0.239
Pneumonia	36 (33.0)	23 (28.7)	0.531
Post-extubation respiratory failure	5 (4.6)	6 (7.5)	0.532
Obesity hypoventilation syndrome	1 (0.9)	1 (1.3)	1.000
Postoperative respiratory failure	1 (0.9)	2 (2.5)	0.575
Comorbidities			
Diabetes mellitus	65 (59.6)	38 (47.5)	0.098
Coronary artery disease	53 (48.6)	33 (41.3)	0.315
Hypertension	49 (45.0)	34 (42.5)	0.737
Chronic kidney disease	6 (5.5)	5 (6.3)	1.000
Other clinical variables			
Home NIV	12 (11.0)	13 (16.3)	0.293
Home long-term oxygen therapy	23 (21.1)	22 (27.5)	0.307
ICU mortality rate	11 (10.1)	25 (31.3)	<0.001
NIV failure in the ICU	43 (39.4)	41 (51.2)	0.107
Age, years	65 (36–86)	65 (45–79)	0.986
CCI, score > 3 points	53 (48.6)	32 (40.0)	0.239
SOFA score	8 (0–14)	10.5 (3–17)	<0.001
APACHE II	18 (15–29)	19 (15-28)	0.089
Glasgow Coma Scale	15 (12–15)	15 (13–15)	0.756
NIV time day	5 (3-9)	7.5 (3–14)	<0.001
Length of ICU stay, day	7 (4–12)	10 (4–18)	<0.001
Laboratory findings			
pH	7.19 (7.05–7.33)	7.19 (7.05–7.33)	0.975
pCO_2_	59 (47–72)	61 (47–75)	0.411
HCO_3_	35 (27–43)	34 (26–42)	0.347
PO_2_	74 (53–88)	74 (60–88)	0.379
Lactate	1.7 (1–2.7)	1.9 (1–3.2)	<0.001
TSH	1.78 (0.18–4.12)	1.89 (0.21–3.9)	0.936
fT3	1.1 (0.1–1.91)	1.25 (0.28–1.93)	0.604
fT4	12.4 (9.2–17)	12.78 (9.35–17)	0.507
CRP	90.09 (1.1–195.8)	120.18 (2.6–198.26)	0.126

ESS, Euthyroid sick syndrome; COPD, Chronic obstructive pulmonary disease; NIV, Non-invasive mechanical ventilation; ICU, Intensive care unit; CCI, Charlson’s comorbidity index; SOFA, Sequential Organ Failure Assessment; APACHE II, Acute Physiology and Chronic Health Evaluation; TSH, thyroid stimulating hormone; CRP, C-Reactive Protein. Data are reported as mean ± SD or percentage *n* (%) or the median (min–max). Continuous and categorical variables were compared between groups with the Mann–Whitney U test or Pearson’s chi-squared test, respectively. *p* < 0.05 was considered statistically significant in all analyses.

**Table 6 medicina-61-00927-t006:** Multivariable analyses that show independently associated factors for ICU mortality.

Parameters	HR	95% CI		*p*-Value
		Lower Limit	Upper Limit	
For the ICU mortality ^1^				
Having the ESS	3.470	1.046	11.512	0.042
APACHE-II scores	1.117	0.996	1.251	0.058
Lactate levels	22.465	5.248	96.169	<0.001
LOS in the ICU, day	1.214	1.043	1.413	0.012
For the NIV failure ^2^				
CCI	0.443	0.198	0.991	0.047
SOFA	1.715	1.462	2.011	<0.001
APACHE-II scores	1.384	1.209	1.585	<0.001
Lactate levels	19.442	4.397	85.970	<0.001
CRP	1.008	1.000	1.016	0.056
LOS in the ICU, day	1.360	1.114	1.660	0.003

^1^ This multivariable binary logistic regression analysis included the variables that were significantly different in the univariate analyses when compared between deceased and surviving patients (having ESS, COPD, home NIV needed, NIV failure, SOFA and APACHE-II scores, lactate and CRP levels and LOS in the ICU). Low fT3 group, ESS subgroups, fT3 and fT4 levels, and NIV time were not included due to the high correlation rate with other parameters. The backward method was used, and the last step (step 6) is shown in the table. The Omnibus test had *p*-values <0.001, and the Nagelkerke R square was 0.557 for this step. ^2^ This multivariable binary logistic regression analysis included the variables that were significantly different in the univariate analyses when compared between patients with and without NIV failure (having ESS, CCI, SOFA and APACHE-II scores, lactate and CRP levels, and LOS in the ICU). Low-fT3 group, fT3, and NIV time were not included due to the high correlation rate with other parameters. The backward method was used, and the last step (step-2) is shown in the table. The Omnibus test had *p*-values <0.001, and the Nagelkerke R square was 0.797 for this step.

**Table 7 medicina-61-00927-t007:** ROC analyses that show the best cut-off values for the thyroid function test levels for predicting ICU mortality and NIV failure.

Parameters	AUC	Cut-off	*p*-Value	Sens. %	Spec. %	PPV %	NPV %
For the ICU mortality							
fT3	0.631	≤1.33	0.002	53.7	71.1	23.2	90.4
fT4	0.608	>11.49	0.009	81.5	36.7	17.3	92.4
TSH	0.518	≤2.78	0.655	90.7	20.8	15.7	93.2
For the NIV failure							
fT3	0.630	<1.77	<0.001	59.6	62.0	46.0	73.8
fT4	0.518	>12.37	0.558	57.4	50.4	38.6	68.5
TSH	0.512	≤3.13	0.701	90.4	16.8	37.2	76.4

Abbreviations: ICU, Intensive care unit; NIV, Non-Invasive Failure; TSH, Thyroid-Stimulating Hormone. Sens.: Sensitivity; Spec.: Specificity; PPV: Positive predictive value; NPV: Negative predictive value.

## Data Availability

The data supporting this study’s findings are available on request from the corresponding author. Due to privacy ethical restrictions, the data are not publicly available.

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
