# Peer review of "Non-Thyroidal Illness Syndrome: A Predictor of Non-Invasive Ventilation Failure and Mortality in Critically Ill Patients"

_medicina, 2025, doi:10.3390/medicina61050927_

Round 1

Reviewer 1 Report

Comments and Suggestions for Authors

This manuscript  investigated the effect of Euthyroid Sick Syndrome on non-invasive mechanical ventilation failure an in-hospital mortality. Results showed that ESS, specifically low  levels of fT3, was significantly associated with increased mortality rates in ICU patients which is associated with higher failure rate of NIV. Study is well designed and implemented. Description of the results are good enough. Drawbacks of the study were clearly stated in the limitations section. I don't have any concerns except cited references. Most of the places references are not cited and instead of keeping three to four citations at the end of four to five sentences, better to cite individually. 

Reviewer 2 Report

Comments and Suggestions for Authors

Thank you for the opportunity to review the manuscript. This study aimed to investigate the effect of Euthyroid Sick Syndrome (ESS) on non-invasive mechanical ventilation (NIV) failure and in-hospital mortality, as well as the impact of prolonged low fT3 levels before ICU admission on these outcomes. The study included 386 ICU patients who received NIV, categorized into two groups: those with ESS and those without (Non-ESS).

The retrospective analysis of fT3 levels in the ESS group over six months before ICU admission identified 61 patients with prolonged ESS.

The results show that ESS was present in 49.0% of the total cohort, with a significantly higher prevalence among deceased patients (66.7%) compared to survivors (46.1%, p=0.005). How accurate are the data? Who collected and how such data are relevant giving the limitation of retrospective studies. Additionally, significant disparities were noted between survivors and deceased patients regarding low fT3 levels (p<0.001), and NIV failure was significantly more common in the ESS group (44.4%) compared to the non-ESS group (26.4%, p<0.001). How can we prove these data are relable giving the large difference comapred to otehr studies in the literature

The primary endpoints—NIV failure and mortality—were compared between the two groups only retrospectively. What is needed for a prospective study. These data are interesting and hupotheiss generating but cannot show definitive results

Major concerns include the need for more clarity in the statistical analysis and potential confounding variables that should be addressed.

Minor concerns involve minor typographical errors and inconsistencies in terminology.

The conclusion highlights that ESS, primarily characterized by low fT3 levels, is significantly associated with increased mortality rates in ICU patients and higher NIV failure rates. Based on these findings, I recommend acceptance of the paper after revisions to improve the clarity and precision of the statistical analysis. Moreover the limitation session should suggest this is a retrospective study to feed a prospective analysis of some sort in the future

Comments on the Quality of English Language

It is already good can be improved because the manuscript is a bit too long.
